# DOES ENTITY ABSTRACTION HELP GENERATIVE TRANSFORMERS REASON?

## ABSTRACT

Pre-trained language models (LMs) often struggle to reason logically or generalize in a compositional fashion. Recent work suggests that incorporating external entity knowledge can improve language models' abilities to reason and generalize. However the effect of explicitly providing entity abstraction remains unclear, especially with recent studies suggesting that pre-trained models already encode some of that knowledge in their parameters. In this work, we study the utility of incorporating entity type abstractions into pre-trained Transformers and test these methods on four different NLP tasks requiring different forms of logical reasoning: (1) compositional language understanding with text-based relational reasoning (CLUTRR), (2) abductive reasoning (ProofWriter), (3) multi-hop question answering (HotpotQA), and (4) conversational question answering (CoQA). We propose and empirically explore three different ways to add such abstraction: (i) as additional input embeddings, (ii) as a separate sequence to encode, and (iii) as an auxiliary prediction task for the model. Overall our analysis demonstrate that models with abstract entity knowledge performs better than without it. However, our experiments also show that the benefits strongly depend on the technique used and the task at hand. The best abstraction aware models achieved an overall accuracy of 88.8% and 91.8% compared to the baseline model achieving 62.3% and 89.8% on CLUTRR and ProofWriter respectively. In addition, abstraction-aware models showed improved compositional generalization in both interpolation and extrapolation settings. However, for HotpotQA and CoQA, we find that F1 scores improve by only 0.5% on average. Our results suggest that the benefits of explicit abstraction is significant in formally defined logical reasoning settings such as CLUTRR and ProofWriter, but point to the notion that explicit abstraction is less beneficial for NLP tasks having less formal logical structure.

## 1 INTRODUCTION

Transformer language models (TLMs; Vaswani et al. 2017) have enabled rapid progress in natural language processing (NLP). When pre-trained on large corpora (such as the web) to predict the next tokens or a set of masked tokens from an input sequence, TLMs can capture linguistic knowledge (Peters et al., 2018; Goldberg, 2019; Tenney et al., 2019b), and yield state-of-the-art performance on many NLP tasks with little to no task supervision (Devlin et al., 2019; Radford et al., 2018; 2019; Brown et al., 2020). However, it is not clear if these models can capture higher level knowledge such as reasoning skills that can be re-used in arbitrary contexts, and in ways that leverage the compositionality of those skills (Lake & Baroni, 2018; Liška et al., 2018), something logical reasoners can do relatively well on a smaller scale (De Raedt et al., 2007; Fierens et al., 2015). Simple compositional tasks such as SCAN (Lake & Baroni, 2018), CLUTRR (Sinha et al., 2019), and ProofWritter (Clark et al., 2020; Tafjord et al., 2020) can help diagnose the compositional generalization behavior of language models. Recent work on some of these datasets showed that TLMs still struggle to learn reasoning strategies that can be re-used in out-of training distribution settings (Lake & Baroni, 2018; Gontier et al., 2020).

If we look at how logical reasoners operate, we find that they have an important abstraction component (going from grounded entities to higher level concepts) before logical reasoning can start (De Raedt et al., 2007). Going from an original text sequence to its higher-order meaning is an important part of the NLP pipeline (part of it being entity type tagging). Similarly in mathematics, the introduction of generic variables allows to progress in a logical reasoning process without keeping track of every

(grounded) atomic entity. Overall, this idea that we call abstraction, seems to be an important part of logical reasoning. Recent work suggests that incorporating external knowledge about grounded entities could improve language models' abilities to reason and generalize (Ji et al., 2017; Zhang et al., 2019; Moosavi et al., 2020; Rosset et al., 2020). However the empirical effect of incorporating generic entity types remains unclear, especially with recent studies suggesting that pre-trained models already encode some of that linguistic knowledge in their parameters (Hewitt & Manning, 2019; Liu et al., 2019a; Clark et al., 2019; Tenney et al., 2019a). In this work, we study the effect of explicitly providing entity type abstraction *in addition to* the original input to pre-trained Transformers.

We explore and evaluate different ways to incorporate entity type abstraction and observe that some methods are more efficient than others. To constructs the abstract representation incorporated into TLMs, we leverage symbolic NLP representations such as entity type information given by popular NLP libraries. This allows for automatic and reproducible data processing. In general, our approach is the following: given an input sequence, we build a simplified version of it by replacing entity names by their corresponding entity types. This simplified sequence can then be used as extra input (Sections 3.1 & 3.2) or as extra training signal to the model (Sections 3.3). In particular, we explore three different ways to augment pre-trained Transformers with this abstract knowledge:

1. by combining token embeddings from both the original and the abstract sequence before encoding (Section 3.1) (Figure 1a & 1b).
2. by encoding both the original and the abstract sequence and combining them before decoding the target output (Section 3.2) (Figure 1c & 1d).
3. by adding a second language model head on top of the Transformer decoder to predict the abstract sequence (Section 3.3) (Figure 1e).

A series of controlled experiments on two synthetic datasets show that models having access to abstract knowledge about entity types yield better performance at inference time both when interpolating and extrapolating to unseen lengths of reasoning chains. Synthetic data is used in order to control for the degree of compositional generalization required. Furthermore, in order to understand if the benefits observed in the synthetic cases are applicable in more realistic settings, we ran a series of experiments on two question answering datasets requiring various degrees of multi-hop reasoning. Unfortunately, results on these more natural language datasets show that abstraction aware models are not significantly better than baseline models.

Overall our work contributions are the following:

1. we introduce and compare empirically different ways to incorporate abstraction into pre-trained TLMs.
2. we show that incorporating abstract knowledge can significantly improve reasoning and compositional generalization in both interpolation and extrapolation when the environment is formally defined in a logical reasoning setting.
3. we show that abstraction aware models may not benefit much when language is more natural and less procedural.

We hope that our work will inspire future research in the field to look for simple inductive biases that can complement pre-trained models in their quest to achieve logical reasoning at scale.

## 2 Related Work

Augmenting neural language models with knowledge about entities has been a popular method to improve their functionality. Ji et al. (2017) trained an entity neural language model to predict sequences of entities with an LSTM (Hochreiter & Schmidhuber, 1997). At each sampling step, they predict the next word alongside a categorical variable indicating the current token's entity ID. They obtained lower perplexity and better results on co-reference resolution and entity prediction tasks than a variety of baselines. Similarly, Rosset et al. (2020) trained a GPT2 model (Radford et al., 2019) by giving it access to entity knowledge at the input level and as an additional pre-training loss. Their model achieved better factual correctness on benchmarks such as LAMA (Petroni et al., 2019), and performed better than a baseline GPT2 model in various question answering tasks. Inspired by this work and motivated by the goal of building better reasoning language models, we instead focus on the prediction of entity *types* rather than entity *identifiers* taken from a fixed list of entities. This

allows our solution to be robust to new entities. In addition, we explore and compare different ways to incorporate the entity knowledge in an encoder-decoder architecture.

Besides entity knowledge, other types of explicit information has also been given to Transformer models. Levine et al. (2019) trained a BERT-like model to learn word senses. They gave their model access to WordNet supersenses at the input level and as an additional training loss. They achieve better performance than other baselines on the SemEval Word Sense Disambiguation task (Raganato et al., 2017). In addition, Moosavi et al. (2020) propose to improve robustness to data biases by augmenting the training data with predicate-argument structures. They train a BERT-base model (Devlin et al., 2019) with PropBank-style semantic role labeling (Shi & Lin, 2019) on MultiNLI (Williams et al., 2017) and SWAG (Zellers et al., 2018) datasets. Their results show that incorporating predicate-argument structure in the input sequence (only during training) makes the model more robust to adversarial examples in MultiNLI. Furthermore, Sachan et al. (2020) ask if syntax trees can help Transformers to better extract information. They augmented a pre-trained BERT model with a syntax graph neural network in order to encode syntax trees and measure the performance of their model against a BERT model on various tasks. Their results showed that the quality of the trees are highly tied to the performance boost observed. More recently, Porada et al. (2021) extended a RoBERTa model (Liu et al., 2019b) with hypernym abstraction based on WordNet to evaluate the plausibility of events. Their model is able to better predict human plausibility judgement than other RoBERTa baselines. **Although different in application, all these prior works leverage the general idea of explicitly giving more abstract knowledge to Transformer models, hence showing how flexible and generic this strategy can be.** We take a similar approach with entity types, but in the hope of improving the reasoning skills of our baseline model.

A flurry of recent work has also examined ways to augment TLMs with entities from external knowledge bases (Zhang et al., 2019; Peters et al., 2019; Févry et al., 2020; Verga et al., 2020). However, most of the time, these solutions rely on external components such as knowledge graphs with pre-trained entity embeddings, and/or an additional memory. While they often use entity linking as a way to perform co-reference resolution, they do not incorporate higher level of abstractions such as entity types like we do here.

## 3 Introducing Abstraction inductive Biases

In this section we describe five different ways to incorporate abstraction into a pre-trained encoder-decoder model. Given an input sequence $X$, we use existing NLP tools such as `spacy` named entity tagger[1] to make a simplified copy $X_s$ of the input. This is a more generic copy of $X$.

We run the `spacy` recognizer on $X$ to extract entity tags such as *PERSON*, *ORG*, *GPE*, etc... For each entity type, we create $n$ additional vocabulary entries (with randomly initialized embeddings) such as $[\text{PERSON\_1}, \ldots, \text{PERSON\_n}, \text{ORG\_1}, \ldots, \text{ORG\_n}, \ldots]$. Every token in $X$ is then replaced by their (randomly numbered) entity tag to make the simplified sequence $X_s$. If the same entity is present multiple times in $X$, each occurrence will be replaced by the same entity tag in $X_s$. If no entity is found for a token in $X$, the original token's text is kept in $X_s$ (*e.g.* "*Bob Smith has a cat that he loves. Bob also loves Alexandra.*" would be transformed into "*PERSON_11 has a cat that he loves. PERSON_11 also loves PERSON_23.*").

The hyper-parameter $n$ is set such that each *distinct* entity within the same sequence gets a different ID. If the same entity appears more than once in a single example, it will get the same ID every time within that sequence. However, we re-use the same set of entity tags across different examples. The value of $n$ depends on each dataset, details can be seen in Appendix A. Although this may look similar to attributing different entity IDs to each entity regardless of their type, we believe that after seeing many examples, all IDs of the same type will have a similar embedding since we randomly select such ID across examples and across epochs. The number of overlapping entity tags across the entire dataset during many epochs will be large, resulting in pushing the embedding of same-type entity tokens closer together.

To test the effect of having multiple tags for the same entity type, we also ran experiments in which we set $n = 1$ thus forcing overlapping entity tags within each example. However we didn't notice any concluding difference, so the rest of this work will focus on the $n > 1$ setting described above.

---

[1] https://spacy.io/models/en

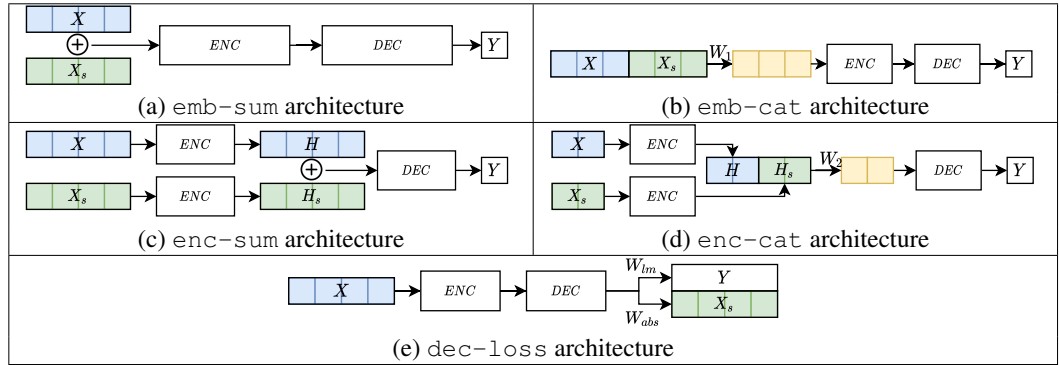

Table 1: Different architectures for different abstraction strategies. $X$ (blue) is the original sequence embedding, $X_s$ (green) is the embedding of the simplified sequence with entities replaced by their entity type tags, "*ENC*" is the T5 encoder, $H$ (blue) is the contextualized representation of sequence $X$, $H_s$ (green) is the contextualized representation of sequence $X_s$, "*DEC*" is the T5 decoder, and $Y$ is the target sequence to predict.

In the following subsections, we describe different strategies to incorporate $X_s$ into an encoder-decoder Transformer model.

## 3.1 Abstraction as an additional embedding

Our first strategy is to combine $X_s$ with $X$ at the embedding level. To do that, we construct $X_s$ to be of the same length of $X$. If a `spacy` entity spans over multiple tokens (*e.g.* ["*Alex*", "*andra*", "*Smith*" "*is*", "*the*", "*wife*", "*of*", "*Bob*"]), we copy its entity tag at each sub-token positions (*e.g.* ["*PERSON_23*", "*PERSON_23*", "*PERSON_23*", "*is*", "*the*", "*wife*", "*of*", "*PERSON_11*"]). For each token within both sequences we either sum (`emb-sum` experiments) or concatenate (`emb-cat` experiments) their respective token embeddings.

**sum**. In `emb-sum` experiments (Figure 1a), if tokens do not have entity tags associated with them, we ignore their embedding to avoid summing the same embedding twice for non-entity tokens. We only sum embeddings of tokens that do have an abstract tag associated with them. This is to ensure that we only modify pre-trained embeddings that correspond to entity tokens. This is done by masking out the tokens in $X_s$ that are the same in $X$. The input given to the model's encoder is then $emb(X) + mask \times emb(X_s) + positional$ with $emb()$ being the embedding matrix, $mask$ defined as the $X \neq X_s$ binary tensor, and $positional$ being the regular Transformer positional embedding. This resembles the setting used by Rosset et al. (2020), however their knowledge-aware embedding comes from a sequence of entities from a dictionary lookup, rather than a sequence of entity *types* from a pre-trained tagger. The advantage of our method is that it is robust to unseen entities of the same type.

**concat**. In `emb-cat` experiments (Figure 1b), if tokens do not have an entity tag associated with them, we replace their embedding with a generic (learnable) "$<grounded>$" token embedding. This ensures that all non-entity token embeddings gets modified in the same way compared to entity tokens. We eventually resize the concatenated embeddings with a learnable matrix $W_1 \in R^{2e \times e}$ with $e$ being the model's embedding size. The input given to the model's encoder is then $[emb(X); emb(X_s)] \cdot W_1 + positional$ with $emb()$ being the embedding matrix, $[;]$ defined as the concatenation operator, and $positional$ being the regular Transformer positional embedding. In this setup the model has $2e^2 + e$ more parameters.

## 3.2 Abstraction as an additional sequence to encode

Our second strategy is to combine $X_s$ with $X$ at the encoding level. To do that, we again construct $X_s$ to be of the same length of $X$. Similarly as above, if a `spacy` entity spans over multiple tokens, we copy its entity tag at each sub-token positions. We then encode both $X$ and $X_s$ with the same encoder weights to have two contextualized encodings: $H$ and $H_s$ respectively. For each token within both sequences we either sum (`enc-sum` experiments) or concatenate (`enc-cat` experiments) their respective contextualized encodings.

**sum**. For `enc-sum` experiments (Figure 1c), the input given to the model's decoder becomes $H + H_s$. Unlike in Section 3.1, in these experiments we do not mask any position because the encodings are contextualized over the entire sequence. All token representations were influenced by all other tokens because of the Transformer encoder attention mask. Thus even non-entity token representations were influenced by entity tokens.

**concat**. For `enc-cat` experiments (Figure 1d), we introduce a learnable matrix $W_2 \in R^{2d \times d}$ with $d$ being the model's encoding size in order to resize the concatenated encodings, similarly to Section 3.1. The input given to the model's decoder is then $[H; H_s] \cdot W_2$ with $[;]$ defined as the concatenation operator. In this setup the model has $2d^2$ more parameters.

### 3.3 ABSTRACTION AS AN AUXILIARY TASK

Our third strategy is to incorporate $X_s$ into the model with an additional cross entropy loss (`dec-loss` experiments). The model will now be tasked to predict both the target output $Y$ as well as the abstracted input $X_s$. To do that, we introduce a second language model head $W_{abs} \in R^{d \times vocab}$ (with $d$ being the model's decoder size) on top of the model's decoder, initialized to have the same weights than the original language model head $W_{lm}$, and fine-tuned during training. We then apply a softmax on the predicted logits before measuring the cross entropy loss between the predicted probability distribution and $X_s$. The final model's loss is then the average between the two cross entropy losses:

$$0.5 * \text{CE}(X_s, \text{softmax}(H^{dec} \cdot W_{abs})) + 0.5 * \text{CE}(Y, \text{softmax}(H^{dec} \cdot W_{lm})),$$

with CE() being the cross entropy loss and $H^{dec}$ being the output tensor of the model's decoder. By sharing the decoder weights (except for the additional language model head), we make sure that most of the network parameters are influenced by the additional cross-entropy loss. This acts as a regularizer and forces the model to "*know*" about entity types within its original parameters. In this setup, the model has $d * vocab$ more parameters. Figure 1e illustrates this strategy.

## 4 EXPERIMENTS

In this section we describe the experiments we ran on various datasets. We start with the CLUTRR benchmark (Sinha et al., 2019) in Section 4.1 as controlled experiments in which we know how much compositional generalisation is required. We then test our models on the ProofWriter (Clark et al., 2020; Tafjord et al., 2020) dataset in Section 4.2. This allows to verify if entity type abstraction is beneficial in formally defined logical reasoning environments with simple language. We next report experiments on the multi-hop question answering task HotpotQA (Yang et al., 2018) in Section 4.3. This allows to test if entity type abstraction is beneficial in two-hop question answering settings with more natural language which is by nature more noisy. Eventually, we also report results on the conversational question answering task CoQA (Reddy et al., 2019) in Section 4.4. This allows to test if entity type abstraction is beneficial in conversational settings in which the entity being discussed may be originally introduced much earlier in the conversation, thus requiring some entity linking before answering.

For all experiments we trained one model for each of the 5 different strategies presented in Section 3, in addition to a baseline model fine-tuned without any abstraction knowledge. We used the `AllenNLP` library (Gardner et al., 2017) with the HuggingFace `transformers` library (Wolf et al., 2019) PyTorch implementation of `T5-small` with 16-bit floating point precision. Each experiment was run on tesla V100 32gb GPUs with early stopping and a patience of 10 epochs on the validation set (defined as a 10% split of the training set). We report all hyper-parameters and library versions in Appendix A for reproducibility purposes.

### 4.1 COMPOSITIONAL GENERALIZATION WITH CLUTRR

Although synthetic, CLUTRR is used because it allows for controlled experiments in which we can clearly measure both interpolation and extrapolation performance of our model. We generated $390,000$ examples that were roughly split 70/30 between training and testing. Each example consists of a unique (non-cyclic) family graph. The goal of this task is to infer the type of edge (family relation) between two nodes (two entities) that are the further apart in the input graph. The bigger the

| CLUTRR | 2 | 3 | 4 | 5 | 6 | 7 | 8 | 9 | 10 | avg. |
|---|---|---|---|---|---|---|---|---|---|---|
| no abstraction | 100% | 84.4% | 64.8% | 61.5% | 54.2% | 56.6% | 45.5% | 42.6% | 56.8% | **62.9%** |
| emb-sum | 100% | 85.3% | 74.6% | 59.3% | 64.3% | 67.7% | 65.1% | 58.9% | 68.4% | **71.5%** |
| emb-cat | 94.6% | 60.0% | 35.6% | 42.0% | 40.2% | 74.3% | 78.2% | 77.9% | 80.6 | **64.8%** |
| enc-sum | **100%** | **94.8%** | **86.9%** | **89.9%** | **85.6%** | **87.2%** | **85.1%** | **84.3%** | **85.5%** | **88.8%** |
| enc-cat | 100% | 86.1% | 72.6% | 70.2% | 66.7% | 69.1% | 63.8% | 61.9% | 74.0% | **73.8%** |
| dec-loss | 100% | 74.7% | 64.9% | 59.3% | 56.2% | 61.3% | 52.8% | 47.8% | 61.3% | **64.2%** |

Table 2: Prediction accuracy on CLUTRR test set for all difficulty levels. Models have been trained on levels 2, 4, 6 with only 9.58% of all the $(e1, rel, e2)$ triples present in the test set. Per-level performance is colored in shades of green for better visualisation.

graph, the more hops are required to infer the missing edge. We express each family graph, along with its question and answer in text using a simple "*[e_1] is the [rel] of [e_2]*" template. Some examples of input/output sequence pairs can be seen in Appendix B.

We evaluate the generated answer accuracy. The answer is defined as the first sentence in the generated sequence. Since all answers during training were expressed using a simple template, we inverse this template to extract the $(\hat{e}1, \hat{rel}, \hat{e}2)$ triple from the generated answer. If the extraction fails, we consider the generated answer wrong. We then compare the extracted triple to the ground truth provided by the CLUTRR dataset. If the reference solution is $(e1, rel, e2)$, we accept both $(e1, rel, e2)$ and $(e2, inv\_rel, e1)$ as valid solutions, with $inv\_rel$ being the inverse relation of $rel$. For instance, the inverse relation of "*father*" can be "*son*" or "*daughter*", we accept both.

### 4.1.1 Testing Compositional generalization

We carefully divided train and test sets to force the model to generalize to both unseen graph sizes (*i.e.*: unseen reasoning depth) and unseen $(e1, rel, e2)$ triples. Specifically, the training set is made of graphs with 2, 4 and 6 relations between 3, 5 and 7 entities respectively, while the test set is made of graphs with up to 10 relations between 11 entities. In addition, all possible entities and relations are seen during training but only 9.58% of $(e1, rel, e2)$ triples from the test set are also in the training set. This small overlap makes the test set harder than originally designed and allows to analyse the compositional generalization capacity of our model.

We fine-tuned a `T5-small` model on $300,000$ training examples of levels $2, 4$ and $6$ and evaluate the model on a test set of $10,000$ examples for all levels from 2 to 10. The level is defined as the number of edges (relations) in the graph. The higher the level is, the bigger the graph is, the further apart the two entities to link are, and the greater the number of hops required to answer the query is. Specifically, we test levels $2, 4$ and $6$ for compositional generalization, levels 3 and 5 for interpolation, and levels $7, 8, 9$ and $10$ for extrapolation.

### 4.1.2 Results

Table 2 shows answer accuracy on each test set level for models trained with $n = 20$ tokens per entity type. As mentioned in Section 3, this helps keeping track of "who's who" in the abstracted sequence, which is important for solving a task such as CLUTRR. We see that the best model (`enc-sum`) strongly outperforms all other models with an average score of 88.8% compared to 62.9% for the baseline.

**sum -vs- concatenation**. We can see that for both the embedding strategies (`emb-cat` & `emb-sum`) and the encoding strategies (`enc-cat` & `enc-sum`), summing representations together yields better performance than feeding their concatenation through a linear layer. One hypothesis is that in the `emb-cat` model, all pre-trained embeddings are modified by either a entity type token embedding or the "<grounded>" embedding, whereas the `emb-sum` model keeps most of its pre-trained embeddings unchanged. This result also suggests that sometimes, simpler methods are more effective.

**embedding -vs- encoding**. From Table 2 we also see that, on average, processing the abstract sequence through the encoder yields better performance than only processing it through the token embedder. This is expected as more layers of the Transformer can process the abstracted sequence.

| ProofWriter | RoBERTa-large (Clark et al., 2020) CWA | no abstraction OWA | emb-sum OWA | emb-cat OWA | enc-sum OWA | enc-cat OWA | dec-loss OWA |
|---|---|---|---|---|---|---|---|
| **Overall** | 83.9% | 89.8% | 90.9% | 90.9% | 88.4% | 90.1% | **91.8%** |
| **D0** | **100.0%** | 99.5% | 99.2% | 99.5% | 98.9% | 99.2% | 99.4% |
| **D1** | **98.8%** | 95.6% | 93.5% | 95.3% | 89.5% | 91.8% | 95.1% |
| **D2** | **98.8%** | 87.9% | 81.0% | 87.1% | 75.3% | 83.5% | 86.6% |
| **D3** | 71.1% | 83.7% | 84.6% | 85.9% | 81.7% | 85.2% | **87.4%** |
| **D4** | 43.4% | 77.3% | **87.4%** | 82.2% | 84.7% | 84.1% | 85.0% |
| **D5** | 37.2% | 70.0% | **85.3%** | 74.8% | 84.0% | 79.6% | 80.6% |

Table 3: Prediction accuracy on different slices of the ProofWriter D5 test set for all our models and the originally reported numbers by Clark et al. (2020). Models have been trained on depth D0, D1, D2. Models are trained in the "open-world" assumption (OWA), except for the original Clark et al. (2020) model which was trained in the "closed-world" assumption (CWA). Per-depth performance is colored in shades of green for better visualisation. The red boxed area indicates test problems at depths unseen during training.

**learning the abstraction**. In the last line of Table 2 the abstraction is given as output to the model. This experiment tests if learning how to abstract along-side the original task helps the model. We can see that learning to predict the abstract sequence can indeed help generalize as the average score increases from 62.9% for the baseline to 64.2% for the `dec-loss` model. However it is not the most effective method in this case.

Overall, we can see that abstraction-aware models can all extrapolate better than the baseline model, suggesting that entity abstraction does help pre-trained Transformers to compositionally generalize. To verify that this is not just a feature of the CLUTRR dataset, we perform the same analysis on another synthetic dataset in the next section.

## 4.2 Abductive Reasoning with ProofWriter

Similarly to CLUTRR, the ProofWriter dataset (Tafjord et al., 2020) is a collection of synthetic facts and rules with derived conclusions. The goal of this task is to infer the truth value of an unknown statement given a series of known facts and 1-hop inference rules. Each example is made of a different set of facts, rules, and an unknown statement. The model then has to predict if the unknown statement is "*True*", "*False*", or "*Unknown*" according to the input knowledge. Each fact and rule is expressed in simple templated language given by a grammar. Some examples of input/output pairs can be seen in Appendix B.

The dataset also provides the required chain of inference required to arrive at the final answer. The number of such 1-hop inference steps is considered the "depth" of the example. For instance a depth-0 (D0) example simply requires to see if the unknown statement is present in the input list of fact or not; a depth-1 (D1) example requires to apply one inference rule to one fact to arrive at the answer; etc... The dataset contains examples of up to depth-5 (D5) inference chains. We test for compositional generalisation by training on examples of up to depth 2 reasoning chains (D0, D1, D2) and testing on examples for each depth from D0 to D5.

We fine-tuned `T5-small` models on the official training and development set from the depth <= 2 data folder and tested it on the test set from the depth <= 5 data folder; consisting of 70,076 training examples and 20,030 testing examples. We trained one model for each of the 5 different strategies presented in Section 3, in addition to a baseline model fine-tuned without any abstraction knowledge. Unlike in all other experiments, for ProofWriter examples, we did not use the generic `spacy` named entity tagger because it did not support the entity types covered by ProofWriter examples. Instead, we used the real abstraction labels provided by the grammar files[2], and defined the following abstraction tokens: "*PERSON*", "*ATTRIBUTE*", "*ANIMAL*", "*RELATION*".

Table 3 shows the prediction accuracy on different slices of the D5 test set for all our models. Although trained on an older version of the dataset ("closed-world" assumption - CWA), we also report the original performance by Clark et al. (2020). While not directly comparable because of the different version of the dataset, we can still see that our `T5-small` experiments all extrapolate (D3-D5) better than the original `RoBERTa-large` model. However `RoBERTa-large` performs

---

[2]https://github.com/allenai/ruletaker/tree/master/grammars_and_config/grammars

|  | HOTPOTQA | | | | CoQA | |
|---|---|---|---|---|---|---|
|  | EM | F1 | Prec | Rec | EM | F1 |
| **no abstraction** | 53.2% | 67.4% | 70.9% | 67.6% | 66.0% | 74.4% |
| **emb-sum** | 52.9% | 67.2% | 70.7% | 67.4% | 65.7% | 74.1% |
| **emb-cat** | 50.9% | 64.9% | 68.2% | 65.3% | 63.9% | 72.4% |
| **enc-sum** | 53.1% | 67.3% | 70.8% | 67.5% | 65.8% | 73.9% |
| **enc-cat** | 51.5% | 65.8% | 69.6% | 65.8% | 65.1% | 73.7% |
| **dec-loss** | **53.7%** | **68.1%** | **71.7%** | **68.2%** | **66.4%** | **74.8%** |

Table 4: Test results for all models. (left) Exact Match, F1, Precision and Recall on HotpotQA. (right) Exact Match and F1 on CoQA.

better at depths seen during training, which may be due to the model size being bigger, hence having greater capacity to model questions of previously seen depths.

Most importantly, if we compare with our "no abstraction" baseline model, Table 3 also shows that our abstraction methods help extrapolate to unseen reasoning depths. While our baseline model performs 83.7%, 77.3%, and 70% on examples from D3, D4, D5 respectively; all other abstraction-aware models extrapolate better, with the best overall model (`dec-loss`) performing 87.4%, 85%, and 80.6% respectively on D3, D4, D5 examples.

Overall, we achieve new state-of-the-art results on the ProofWriter dataset when trained only on examples from D0-D2. This is suggesting that entity abstraction does help pre-trained Transformers to compositionally generalize to unseen reasoning chains. One important thing to note however is that both CLUTRR and ProofWriter sentences are relatively simple to abstract (we found that 94% of CLUTRR entities were correctly labeled as PERSON by `spacy` entity tagger, and 100% of ProofWriter entities are correctly abstracted since we used the golden grammar). In the next sections we will analyse what happens on more realistic data.

### 4.3 MULTI-HOP QUESTION ANSWERING WITH HOTPOTQA

In this section we report experiments on the multi-hop question answering (HotpotQA) dataset (Yang et al., 2018). HotpotQA contains natural language, making it more diverse and harder to get abstraction labels than CLUTRR. In addition, HotpotQA has 2-hop inference chains both in its training and testing data splits. While not as "reasoning intensive" as CLUTRR or ProofWriter, it is still a good compromise between natural language and multi-hop reasoning.

Used in the distractor setting, each example consists of a list of 10 Wikipedia paragraphs, a question that requires the model to combine information from two paragraphs, and the answer. Since concatenating all of the 10 paragraphs would result in a context size much larger than what regular Transformer models allowed (512 or 1024 tokens), we instead only took the two golden paragraphs as context, plus the question. While this beats the original purpose of retrieving the useful paragraphs, we are not interested in achieving state-of-the art on this benchmark. We are rather interested in using it to compare the usefulness of our approach on a more natural multi-hop question-answering setup. An example of input/output sequence pair can be seen in Appendix B.

Because the official test set is not public, we used the official validation set as our test set to compare our models and fine-tuned a `T5-small` model on 90% of the training set while keeping the remaining 10% as our custom validation set for early stopping. We trained one model for each of the 5 different strategies presented in Section 3, in addition to a baseline model fine-tuned without any abstraction knowledge.

Table 4 (left) shows exact match (EM), F1 scores, Precision and Recall on our test set. We can see from these results that the best model is the abstraction-aware `dec-loss` model (trained to predict both the answer and the input in its abstract form) with an F1 score of 68.1% against 67.4% for the baseline model. However, the baseline model is a strong candidate and the abstraction does not always benefit the model depending on how it is incorporated. This may be due to the fact that entity abstraction labels are harder to predict and that entity type abstraction may not be required for some examples. Eventually, we find that `emb-cat` is the least performing model and `enc-sum` is one of the better performing abstraction-aware model (after `dec-loss` in this case).

## 4.4 Conversational Question Answering with CoQA

Eventually, motivated by the use of a generative model, we test the same abstraction strategy in a conversational setting. For that, we leveraged the conversational question answering dataset CoQA (Reddy et al., 2019). The task presented by this dataset is to understand a text passage and answer a series of inter-connected questions in a conversation. The conversation aspect introduces follow-up questions that forces the model to keep track of what entity is currently being referred to and to look back at previous interactions. Examples of input/output sequence pairs can be found in Appendix B. The dataset does not always explicitly forces multi-hop reasoning steps, but it could still happen (*i.e.* see the last CoQA example of Appendix B in which the model must perform a substraction between all subjects and the ones already mentioned). In addition, the conversational nature of this dataset often introduces a co-reference step to be made before fetching the information from the paragraph in context. In this setting we will thus test if abstraction can help in this multi-step information retrieval procedure.

Similarly to HotpotQA, because the official test set is not public, we used the official validation set as our test set to compare our models and fine-tuned a `T5-small` model on 90% of the training set while keeping the remaining 10% as our custom validation set for early stopping. We trained one model for each of the 5 different strategies presented in Section 3, in addition to a baseline model fine-tuned without any abstraction knowledge.

Table 4 (right) shows the exact match (EM) and F1 score on our test set. Although by a small margin, we can see that the best model is again the abstraction aware `dec-loss` model (trained to predict both the answer and the input in its abstract form) with an F1 score of 74.8% against 74.4% for the baseline. The baseline model is quite strong already and the benefit of abstraction is questionable in this case. As in previous experiments, the worst performing model is `emb-cat` and one of the best is the `enc-sum` model (after `dec-loss` in this case).

## 5 Discussion and Future Work

We presented various ways to incorporate abstract knowledge into Transformer Language Models. Focusing on entity types, this work evaluated model performance on reasoning tasks requiring compositional generalization and multi-hop reasoning.

Overall our results demonstrate three things: (i) incorporating abstract knowledge can significantly improve reasoning and compositional generalization in both interpolation and extrapolation when the environment is formally defined in a logical reasoning setting; (ii) different ways to incorporate abstraction can yield different performance boosts: `enc-sum` and `dec-loss` are generally performing better than others; (iii) abstraction aware models may not benefit much when language is more natural and less procedural. This last result suggest that the amount of information provided by off-the-shelf entity recognizers are already captured by pre-trained weights of Transformers. However, the fact that abstraction labels greatly improve generalization in CLUTRR and ProofWriter experiments show that in order for pre-trained models to benefit from abstraction, the sequences must be easy to abstract and the task must have a formal logical structure.

**Limitations**. Along with the fact that abstraction may not help in more realistic data, another limitation of our work is that the method we present requires more annotated data, which is not always available. Furthermore, this additional data processing can take time and may not scale well to larger datasets if implemented naively.

**Future Explorations**. One hypothesis that results from our work is the following question: could *pre-training* Transformer models with additional abstraction data result in stronger performance when fine-tuned with abstraction data like we do in this work? Although we could not train from-scratch a T5 model on its original C4 dataset (Raffel et al., 2020), we believe that augmenting C4 with abstract annotations like we do on a smaller scale and training T5 from scratch on this augmented dataset could potentially yield a stronger language model.

ETHICAL CONCERNS

Previous work showed that pre-trained language models can have some undesired societal biases (Henderson et al., 2018; Shwartz et al., 2020; Bender et al., 2021). Although not explored in this work, we believe that giving abstract entity types like we do in this work could have a positive societal impact on language models, potentially alleviating some of these biases. For instance, through abstraction, a model can be exposed to both female and male names both being "*PERSON*"s and that a "*PERSON*" can equally be a "*manager*" or an "*assistant*" regardless of its gender. Similar exposure could be achieved with other abstraction types such as "*RELIGION*", "*JOB*", "*COUNTRY*", "*NATIONALITY*", etc... One potential future direction for this work could be to measure the impact explicit abstraction can have on benchmarks such as StereoSet (Nadeem et al., 2020).

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

## A  HYPERPARAMETERS

We used the default `T5-small` hyperparameters from the `HuggingFace` library (Wolf et al., 2019). We present in Table 5 below the library version we used and the model hyperparameters used for all experiments.

The only difference between each dataset is the number of same-type entity tags allowed in each input sequence $n$. For CLUTRR the maximum number of same-type entities was 20, for ProofWriter it was 10, and for HotpotQA and CoQA it was 100. This is due to the larger context size and the diversity of natural language texts.

| | |
|---:|:---:|
| $n$ for CLUTRR | 20 |
| $n$ for ProofWriter | 10 |
| $n$ for HotpotQA | 100 |
| $n$ for CoQA | 100 |
| AllenNLP | version 2.2.0 |
| Transformers | version 4.4.2 |
| Spacy | version 2.3.5 |
| batch size | 256 |
| 16-bit floating point | True |
| dim embedding | 512 |
| dim feedforward | 2048 |
| dim key-value | 64 |
| dropout | 0.1 |
| max length | 512 |
| #of heads | 8 |
| #of layers | 6 |
| optimizer | AdamW |
| learning rate | 1.00E-05 |
| betas | [0.9, 0.999] |
| epsilon | 1.00E-08 |
| gradient norm | 1.0 |
| sampler | top-p |
| p | 0.9 |
| temperature | 1.0 |

Table 5: Library version and model hyper-parameters.

## B    INPUT - OUTPUT EXAMPLES

**CLUTRR**.

| lvl.2 input: | ``question :  How is Anne related to Gary ?  context : Brett is Anne 's father .  Gary is a son to Brett .'' |
|---|---|
| output: | ``answer :  Anne has a brother named Gary .'' |
| lvl.4 input: | ``question:  What is the family connection between Patricia and Timothy ?  context :  May is the aunt of Doris .  Patricia has a daughter called Doris .  Timothy is Charles 's brother .'' May has a son called Charles . |
| output: | ``answer :  Timothy is the nephew of Patricia .'' |

**ProofWriter**.

| Depth-0 input: | ``context :  The cow is round.  The cow needs the lion.  The cow needs the rabbit.  The cow sees the lion.  The cow visits the rabbit. The lion is round.  The rabbit is kind.  The rabbit visits the tiger. The tiger is big.  The tiger is kind.  The tiger sees the rabbit. The tiger visits the rabbit.  If something is kind and it visits the rabbit then it is young.  If something sees the tiger and it visits the lion then it sees the rabbit.  If something is big and young then it sees the lion.  If something visits the rabbit then the rabbit needs the lion.  If something is big then it visits the rabbit.  If something sees the tiger then it is rough.  If something visits the rabbit and it is kind then the rabbit needs the lion.  If something is rough and kind then it visits the lion.  If something needs the lion then it is big.  question :  The tiger visits the rabbit.'' |
|---|---|
| Depth-0 output: | ``answer :  True'' |
| Depth-2 input: | ``context :  Anne is nice.  Charlie is blue.  Charlie is furry. Charlie is green.  Charlie is kind.  Charlie is nice.  Charlie is red.  Fiona is furry.  Fiona is green.  Harry is furry.  Harry is kind.  Harry is nice.  All nice people are rough.  If someone is red and furry then they are blue.  Rough, kind people are furry. If Charlie is furry then Charlie is nice.  All furry people are nice.  If someone is rough then they are kind.  If someone is red and nice then they are blue.  All furry people are red. question :  Harry is not blue.'' |
| Depth-2 output: | ``answer :  False'' |

**HotpotQA**.

| input: | ``question :  Which magazine was started first Arthur's Magazine or First for Women ?  context :  Arthur's Magazine (1844–1846) was an American literary periodical published in Philadelphia in the 19th century.  Edited by T.S. Arthur, it featured work by Edgar A. Poe, J.H. Ingraham, Sarah Josepha Hale, and others.  In May 1846 it was merged into ``Godey's Lady's Book''.  First for Women is a woman's magazine published by Bauer Media Group in the USA. The magazine was started in 1989.  It is based in Englewood Cliffs, New Jersey. In 2011 the circulation of the magazine was 1,310,696 copies.'' |
|---|---|
| output: | ``answer :  Arthur's Magazine'' |

**CoQA**.

| context: | ``context :  The Vatican Apostolic Library, more commonly called the Vatican Library or simply the Vat, is the library of the Holy See, located in Vatican City.  Formally established in 1475, although it is much older, it is one of the oldest libraries in the world and contains one of the most significant collections of historical texts.  It has 75,000 codices from throughout history, as well as 1.1 million printed books, which include some 8,500 incunabula.  The Vatican Library is a research library for history, law, philosophy, science and theology.  The Vatican Library is open to anyone who can document their qualifications and research needs.''  [...]  ``Only a handful of volumes survive from this period, though some are very significant.'' |
|---|---|
| input: | context above + ``question :  When was the Vat formally opened?'' |
| output: | ``answer :  It was formally established in 1475.'' |
| input: | context above + ``question :  When was the Vat formally opened?  answer : It was formally established in 1475.  question :  what is the library for?  answer :  research.  question :  for what subjects?'' |
| output: | ``answer :  history, and law.'' |
| input: | context above + ``question :  When was the Vat formally opened?  answer : It was formally established in 1475.  question :  what is the library for?  answer :  research.  question :  for what subjects?  answer : history, and law.  question :  what else ?'' |
| output: | ``philosophy, science and theology.'' |

