# OpenReview forum: "Does Entity Abstraction Help Generative Transformers Reason?"
_ICLR.cc/2022/Conference — ICLR 2022 Submitted_

### Official Review · Reviewer_oWCK · 2021-11-02

**Correctness:** 4
**Technical Novelty And Significance:** 3
**Empirical Novelty And Significance:** 2
**Recommendation:** 5
**Confidence:** 4

**Main Review:**

**Strengths**:
\
Overall a good paper. It is well presented, and the experimental results are solid.

**Weaknesses**:
\
My major concern is about the experiments. Specifically,
1. There is very minor boost in performance on QA datasets. The goal of the proposed model should be improving real-world data, instead of synthetic data (CLUTTR in this paper). Because improving numbers on synthetic data could be hacky, and not scalable to practical applications.
2. The performance of baselines are far below the current top performers. I understand that the paper is not interested in achieving new state-of-the-art, and the test set used might be different. However, in that case you can not prove that the proposed methods are still beneficial on a model that achieves way better baseline results. An analogy here is that something working under low-resource setting is not necessarily effective when there is adequate access to data.
3. The method of incorporating abstraction is not universal. We’ve seen different methods work for different datasets. If there is a new reasoning dataset, which architecture in Table 1 should people use? Testing all five is very inefficient.

**Style, Typos etc.**
1. In Section 4,1, it is mentioned that “the inverse relation of “father” can be “son” or “daughter”, we accept both”. What does “accept both” mean here? “Ann is the daughter of Brett” and “Ann is the son of Brett” can not be both correct. Right?
2. Also in Section 4.1, “We generated 390,000 examples that were roughly split 70/30 between training and testing”. But in Section 4.1.1, the size of test set becomes 10,000. Which one is correct?
3. The lvl.2 output of CLUTTR in Appendix B does not use the answer template specified in Section 4.
4. TACRED has been mentioned multiple times in the Appendix. How is TACRED used in this work?

**Summary Of The Paper:**

The paper investigates the effect of incorporating entity type abstraction into pre-trained Transformers. To achieve that, the authors have tried five different architectures to build the abstraction aware model. The proposed model is tested on three NLP datasets for reasoning. Empirical results show that entity type abstraction is beneficial in formally defined logical reasoning environments with simple language. While for QA datasets with more natural language, the baseline is already very strong and the improvement of incorporating abstraction is minor.

**Summary Of The Review:**

This paper tries to address an important problem, but the experiments conducted show that the proposed methods are not scalable to more real-world datasets. I do not recommend acceptance of the paper in its current form.

---

> ### Author Response · Authors · 2021-11-15
> **Reply to oWCK**
>
> Dear reviewer oWCK, thank you for taking the time to read our paper and write your feedback.
> We are happy to see that you think the paper is well presented and that experimental results are solid. Hence, we believe that you will be pleased to see that we have added similar results on the ProofWriter dataset confirming that the abstraction technique is indeed useful for tasks requiring compositional generalization (see comment to all above & section 4.2 of the new pdf for more details).
>
> Regarding your main concerns,
>
> 1. We would like to respectfully disagree with the claim that “ _improving numbers on synthetic data could be hacky, and not scalable to practical applications_ ”.
> The motivation of this work is to build Transformer models capable of multi-step logical reasoning. We believe that this _task_ is best captured by synthetic datasets such as CLUTRR and ProofWriter. A practical application of this _task_ can be specific to certain real-world problems but is hardly captured by “realistic” datasets such as HotpotQA and CoQA. This work demonstrates that the method is very effective for reasoning problems requiring longer chains of reasoning than those seen during training.
>
> 2. Since we are not interested in improving the state-of-the-art (as you correctly understood), adding abstraction to more complex models is out of the scope for this current work. We first wanted to understand the behavior on more “classical” models. However, we acknowledge that it is an interesting future direction. In addition, note that we are deeply motivated by finding easy to implement inductive biases rather than continually “adding complexity” to already energy-hungry networks. Nevertheless, we believe that adding more representation power to our models will boost overall performance for all our methods.
>
> 3. We agree that further analysis on how different abstraction techniques influence performance should be done. This is part of our future work road-map. Currently we found that _enc-sum_ and _dec-loss_ architectures seem to be more promising for future research.
>
> Style & Typos:
>
> 1. What we meant by this admittedly confusing statement, is that if the target relation is (Brett, father-of, Ann) we want to accept (Ann, child-of, Brett). However there is no “child-of” relationship in the dataset. So to circumvent this limitation, we indeed take a gender-neutral approach and accept both (Ann, son-of, Brett) and (Ann, daughter-of, Brett). CLUTRR is known to be un-realistic.
>
> 2. In Section 4.1.1 we mention that we “ _evaluate the model on a test set of 10k examples for all levels from 2 to 10_ ”. By this we mean 10k examples for each of the 9 levels, summing up to 90k examples total. We will be sure to clarify this in the final version of the paper.
>
> 3.  " _The lvl.2 output of CLUTTR in Appendix B does not use the answer template specified in Section 4._  "
> That’s because we have 4 to 5 different templates available but they are all as simple as the one mentioned in the main text.
>
> 4. " _TACRED has been mentioned multiple times in the Appendix. How is TACRED used in this work?_ "
> You found the easter egg! We are sorry for the forgotten mistake and caused confusion, this is now replaced with additional results on a new dataset (ProofWriter) that further supports our conclusions.
>
> We hope to have addressed some of your major concerns, if not do not hesitate to reply back to us, it will be a pleasure to continue discussing and make our paper stronger.

---

### Official Review · Reviewer_sWnL · 2021-11-03

**Correctness:** 3
**Technical Novelty And Significance:** 2
**Empirical Novelty And Significance:** 2
**Recommendation:** 6
**Confidence:** 3

**Main Review:**

strengths:

1. This paper is in a good shape for reading, easy following.

2. The experiment part gives convincing numbers on performance boosting on three NLP tasks, CLUTRR, HotpotQA, and CoQA.

weaknesses:

1. The experiment part lacks analysis on the model which could provide more intuitions on how the model works. More analytic experiments will bring more interesting findings of the proposed model. For instance, we could dive deep into why the model variants perform differently between CLUTRR and HotpotQA & CoQA.



**Summary Of The Paper:**

This paper empirically explored three different ways to add abstraction, additional input embeddings, a separate sequence to encode, and an auxiliary prediction task. Experiments on CLUTRR, HotpotQA, and CoQA show that the models with abstract entity knowledge perform slightly better than without it.

**Summary Of The Review:**

Overall, this is a solid empirical paper with convincing experimental results. It will be of great help for revealing the intuition of the model if introducing more analytic experiments.

---

> ### Author Response · Authors · 2021-11-15
> **Reply to sWnL**
>
> Dear reviewer sWnL, thank you for taking the time to read our paper and write your feedback.
> We are happy to see that you think the paper is well written and easy to read. We are also pleased that our results convinced you about the importance of entity type abstraction.
> We believe that you will be happy to see that we have added another set of strong results on the ProofWriter dataset confirming that the abstraction technique is indeed useful for tasks requiring compositional generalization (see comment to all above & section 4.2 of the new pdf for more details).
>
> We agree that further analysis on how different abstraction techniques influence performance should be done. This is part of our future work road-map.
>
> Regarding the reason why abstraction significantly improved CLUTRR (and now ProofWriter) experiments and not the other two “natural language” tasks, we believe that this simply comes from the fact that "real-world" problems likely do not have strong enough logical structure which can benefit from the abstraction technique. Refer to the comment to all reviewers above for further discussion.
>
> Do not hesitate to reply back to us, it will be a pleasure to continue discussing and make our paper stronger.

---

### Official Review · Reviewer_pDcD · 2021-11-03

**Correctness:** 3
**Technical Novelty And Significance:** 2
**Empirical Novelty And Significance:** 2
**Recommendation:** 3
**Confidence:** 4

**Main Review:**

Strengths
- The paper is clearly written and easy to understand.
- The improvement on the synthetic logical reasoning setting (CLUTRR) is significant.
- The conclusion of the paper provides some insights on the capacity of T5-small model to understand entity abstraction.

Weakness
- The paper only shows end-to-end performance on the three tasks without deeper analysis on why entity abstraction does not always improve the performance. Like the authors pointed out, the current conclusion could be due to the noisy entity type predictions or the fact that the tasks do not require entity abstraction. Without the deeper analysis, readers cannot generalize the conclusion beyond the three datasets mentioned in this paper to other unexplored downstream tasks.
- Only one pretrained transformer model t5-small is tested, therefore the empirical findings of this paper cannot be applied to other models, which usually contain much more parameters and perform better on various downstream tasks.

**Summary Of The Paper:**

The paper studied the utility of incorporating entity type abstractions into pre-trained Transformers and its effectiveness on three tasks which require logical reasoning: 1) compositional language understanding with text-based relational
reasoning 2) multi-hop question answering 3) conversational question answering. Empirical result shows that the proposed method significantly improves the synthetic compositional language understanding task while only marginally improves the other two tasks in natural languages.

**Summary Of The Review:**

Technically, the paper presents novel approaches to incorporate entity type abstraction with transformer language model. As the related work section discussed, there are a lot of other related works also incorporating knowledge from named entities. To me, using only the entity type is marginally novel. Also, the empirical findings lack deeper analysis and were merely conducted on a single and toy-sized pretrained language model. For empirical results, the only significant improvement is on a synthetic dataset while the gain on real-world datasets is also marginal. Therefore, the significance of the paper's contribution is limited.

---

> ### Author Response · Authors · 2021-11-15
> **Reply to pDcD**
>
> Dear reviewer pDcD, thank you for taking the time to read our paper and write your feedback.
> We are happy to see that you think the paper is well written and easy to understand. We are also pleased that you recognized the significance of the improvements on logical reasoning with CLUTRR experiments. Hence, we believe that you will be pleased to see that we have added similar results on the ProofWriter dataset confirming that the abstraction technique is indeed useful for tasks requiring compositional generalization (see comment to all above & section 4.2 of the new pdf for more details).
>
> Regarding your main concerns,
>
> - Thank you for pointing out that the conclusion needs more analysis. We will try to express our intuition below, but we are always open to additional experiment suggestions that could show this.
> Since we don’t have access to ground truth abstraction labels on the natural language datasets, it is hard to analyse what could have happened if we did. However the reported “performance” of the entity tagger we used is quite strong (0.85 F1 score [1] -- we will make sure to mention this important metric in the final version of the paper). Thus, we hypothesize that the noisy labels represent _only a small part_ of the reason why it doesn’t work so well. We believe that most of the reason why it doesn’t work on “realistic” datasets simply comes from the fact that "real-world" problems likely do not have strong enough logical structure which can benefit from the abstraction technique.
>
> - While t5-small is the only model tested, we don’t believe it to be “toy-sized”. Our models have between 181M and 194M parameters. Sure it is nothing like a GPT3, but already the amount of resources required to train them on various datasets in various configurations is not available to all. In addition, we are deeply motivated by finding easy to implement inductive biases rather than continually “adding layers” to already energy-hungry networks. Nevertheless, we believe that adding more representation power to our models will boost overall performance for all our methods.
> Furthermore, in our additional experiments we compared performance with a RoBERTa-large model and found that all our t5-small models outperformed the larger model when it comes to longer reasoning chains (see table in the comment to all reviewers above and new Section 4.2 in the updated manuscript). Note that the fact that RoBERTa-large performs better at levels D0, D1, D2 indicates that bigger models can better capture training signals, however, without abstraction, they struggle to generalize to unseen reasoning depths.
>
> We hope to have addressed some of your major concerns, if not do not hesitate to reply back to us, it will be a pleasure to continue discussing and make our paper stronger.
>
> [1] https://github.com/explosion/spacy-models/releases/tag/en_core_web_lg-3.2.0

---

### Official Review · Reviewer_r8rD · 2021-11-05

**Correctness:** 2
**Technical Novelty And Significance:** 2
**Empirical Novelty And Significance:** 1
**Recommendation:** 3
**Confidence:** 4

**Main Review:**

Strengths:
1. The paper is generally well written and easy to follow. The idea of using entity abstraction is simple to implement and can be widely applied if works.
2. The experimental results on CLUTRR is promising and show improved performance on compositional generalization, which is interesting.

Weaknesses:
1. The idea of using entity abstraction is straightforward and not very novel as doing POS tagging for input is widely used in many NLP models, especially in traditional NLP pipelines.
2. While the performance on synthetic datasets is promising, on realistic datasets the proposed method fails to improve the performance. Therefore the effectiveness of the method is not fully supported by the experiments.

**Summary Of The Paper:**

This paper investigate incorporating entity abstraction to transformer language models for text reasoning tasks. The paper proposes different methods to inject entity abstraction information into transformer LMs and experiments on a synthetic dataset show that the proposed method helps compositional generalization. However, experiments on two realistic datasets show that the proposed method fail to effectively improve performance.

**Summary Of The Review:**

This paper proposed a simple method which incorporating entity abstraction to transformer language models. The method is simple while somewhat trivial, and the novelty/technical contribution is not very significant. The experimental results on synthetic dataset is good but not as well on realistic datasets. Therefore I believe this paper is below the bar.

---

> ### Author Response · Authors · 2021-11-15
> **Reply to r8rD**
>
> Dear reviewer r8rD, thank you for taking the time to read our paper and write your feedback.
> We are happy to see that you think the paper is well written and easy to follow. We are also pleased to have triggered your interest regarding the major improvements on compositional generalization with CLUTRR experiments. Hence, we believe that you will be pleased to see that we have added similar results on the ProofWriter dataset confirming that the abstraction technique is indeed useful for tasks requiring compositional generalization (see comment to all above & section 4.2 of the new pdf for more details).
>
> Regarding your main concerns,
>
> 1. While POS tagging for input is widely used in traditional NLP pipelines, it is not the case for traditional Transformer models. These models are usually trained on raw text with the language modeling objective. Our work takes the in-between approach of incorporating a classical NLP pipeline toolkit into a Transformer model.
> Also, we don’t do POS tagging but NER.
> The technique may be simple to implement but, as you noted, “ _the idea of using entity abstraction is simple to implement and can be widely applied if it works_ ”. We see the simplicity of the method as a strength rather than a weakness.
>
> 2. As discussed in the comment to all reviewers above, we agree that the proposed method is not effective on realistic tasks. However this is part of our message. If you have a formal problem that requires various depths of reasoning, abstraction can significantly help at inference time (we added experiments on ProofWriter to further show that - see Section 4.2 in the updated pdf) - but we also took rigorous time to show that informal natural language tasks requiring informal reasoning do not benefit from such techniques. Our results make it clear that this is not the setting it should be used for. That doesn’t mean the method is not useful, it only means that it should not be used in certain situations. Our results show that abstraction should be used (and is indeed beneficial) in formally defined reasoning tasks such as CLUTRR and ProofWriter. We will make sure to clarify that  conclusion in the final version of the paper.
>
> We hope to have addressed some of your major concerns, if not do not hesitate to reply back to us, it will be a pleasure to continue discussing and make our paper stronger.

---

### Author Response · Authors · 2021-11-11
**Reply to all & Update #1: additional experiments**

Thank you all for taking the time to read our paper and write your feedback.

We will address individual concerns in more detail soon, but we wish to introduce new experiments on the ProofWriter dataset [1] and to clarify a common misconception about our results and contribution right away.

We wish to clarify the following critical point. Our work here seeks to determine the utility of adding explicit abstractions to logical reasoning tasks, encoded in natural language. Our results on problems with clear and well defined logical structure show significant gains and moreover, gains of different strength for different methods of adding abstractions. Yes, indeed, when this approach is used for so-called “real-world” problems involving natural language (such as CoQA and HotpotQA) none of the explicit abstraction methods explored here significantly improve results, or differ from one another. However, this is precisely our key point and finding, namely that explicit abstraction appears to help only when there is **formally definable, multistep logical structure** present in the underlying problem. These other “real-world” problem domains involve “reasoning” steps related to natural language, but do _not_ involve formal multi-step logical reasoning in a well defined setting. The CoQA task involves question answering which typically transforms into span prediction combined with the task of coreference resolution, whereas HotpotQA involves combining information within two paragraphs of organically sourced text. We selected these “real-world” datasets and (importantly) their associated tasks as they were among the closest NLP tasks to formal logical inference problems (of which we are aware) and the fact that our methods do not improve results are an important part of our message -- as in ML not all examples should be positive, in science not all results should be positive (incrementally improving various benchmarks).

**Our key result is that our proposed methods dramatically enhance the performance of transformers when tasked with performing multistep logical inference in a well defined setting.**

To further underscore this observation we have run additional results on the Proofwriter task [1], where our interpretation of these results is given further support with a very clear result. We have added details about this experiment and the results in Section 4.2 in the updated pdf. In brief, we trained models on examples involving reasoning of depth up to length 2 and tested them on questions requiring logical reasoning up to depths of 5. Our results show that here again, on this formally defined set of logical inference tasks, explicit abstraction helps models to generalize to longer chains of reasoning than those used for training. This result combined with the results in our original submission strongly support our interpretation that abstraction helps in tasks that require explicit multi-step abstract logical reasoning as exemplified by the CLUTRR and ProofWriter problem domains - but that informal natural language tasks requiring informal reasoning do not benefit from such techniques. We provide the key table of results that we have added to the manuscript below for quick reference.

| ProofWriter | RoBERTa-large [2] | no abstraction | emb-sum | emb-cat | enc-sum | enc-cat | dec-loss |
| ----------- | ---------------------------------- | -------- | ------- | ------- | ------- | ------- | -------- |
| **Overall**     | 83.9%                              | 89.8%    | 90.9%   | 90.9%   | 88.4%   | 90.1%   | **91.8%**    |
| **D0**          | **100.0%**                             | 99.5%    | 99.2%   | 99.5%   | 98.9%   | 99.2%   | 99.4%    |
| **D1**          | **98.8%**                              | 95.6%    | 93.5%   | 95.3%   | 89.5%   | 91.8%   | 95.1%    |
| **D2**          | **98.8%**                              | 87.9%    | 81.0%   | 87.1%   | 75.3%   | 83.5%   | 86.6%    |
| **D3**          | 71.1%                              | 83.7%    | 84.6%   | 85.9%   | 81.7%   | 85.2%   | **87.4%**    |
| **D4**          | 43.4%                              | 77.3%    | **87.4%**   | 82.2%   | 84.7%   | 84.1%   | 85.0%    |
| **D5**          | 37.2%                              | 70.0%    | **85.3%**   | 74.8%   | 84.0%   | 79.6%   | 80.6%    |

[1] Tafjord, Oyvind, Bhavana Dalvi Mishra, and Peter Clark. "Proofwriter: Generating implications, proofs, and abductive statements over natural language." _arXiv preprint arXiv:2012.13048_ (2020).

[2] Clark, Peter, Oyvind Tafjord, and Kyle Richardson. "Transformers as soft reasoners over language." _arXiv preprint arXiv:2002.05867_ (2020).

---

### Author Response · Authors · 2021-11-26
**short message to all**

We think there is potentially a key misunderstanding of the nature of our experimental results leading a number of reviewers to score our paper as a reject. We just want to make sure this key point is clear to everyone. Our main conclusion is that adding explicit abstraction is **not necessary for shallow surface level reasoning tasks** of the type typically encoded within large text processing tasks (even CoQA and HotpotQA); however, explicit abstraction indeed becomes **very useful and important for tasks that require long chains of reasoning**. Furthermore, the way in which such abstractions are given to a neural architecture has a significant impact on performance. **We have added a second set of strong and clear positive results for the Proofwriter task (another task which requires multi-step logical inferences). These results are consistent with our previous study on CLUTRR**.

Thanks for taking some time to read our work.

---

### Decision · Program_Chairs · 2022-01-20

**Decision:**

Reject

**Comment:**

This is a clearly written paper about integration of entity abstraction to the transformer based language modeling methods for language processing tasks that require reasoning (this is clarified by the authors later as tasks that require linger chains of reasoning) and have shown results on CLUTTR, HotpotQA, and CoQA. The reviewers seem to agree on two issues: First, it is not clear why the proposed idea does not result in a lot of improvement, except the synthetic CLUTTR. Authors provided additional experimental results on yet another dataset. Second, the paper would benefit from a detailed analysis of the experimental results, for example, why don't abstractions help on all datasets.